# Kidney Injury by Variants in the *COL4A5* Gene Aggravated by Polymorphisms in Slit Diaphragm Genes Causes Focal Segmental Glomerulosclerosis

**DOI:** 10.3390/ijms20030519

**Published:** 2019-01-26

**Authors:** Jenny Frese, Matthias Kettwig, Hildegard Zappel, Johannes Hofer, Hermann-Josef Gröne, Mato Nagel, Gere Sunder-Plassmann, Renate Kain, Jörg Neuweiler, Oliver Gross

**Affiliations:** 1Clinic of Nephrology and Rheumatology, University Medical Center Goettingen, 37075 Goettingen, Germany; jenny.frese@dpdhl.com; 2Clinic of Pediatrics and Adolescent Medicine, University Medical Center Goettingen, 37075 Goettingen, Germany; matthias.kettwig@med.uni-goettingen.de (M.K.); hzappel@med.uni-goettingen.de (H.Z.); 3Department of Pediatrics, Pediatrics I, Innsbruck Medical University, 6020 Innsbruck, Austria; Johannes.Hofer@i-med.ac.at; 4Department of Cellular and Molecular Pathology, German Cancer Research Center, 69120 Heidelberg, Germany; h.-j.groene@dkfz-heidelberg.de; 5Center for Nephrology and Metabolic Disorders, Molecular Diagnostics, 02943 Weißwasser, Germany; nagel@moldiag.de; 6Division of Nephrology and Dialysis, Department of Medicine III, Medical University of Vienna, 1090 Vienna, Austria; Gere.Sunder-Plassmann@meduniwien.ac.at; 7Department of Pathology, Medical University of Vienna, 1090 Vienna, Austria; renate.kain@meduniwien.ac.at; 8Institute of Pathology, Kantonsspital, 9007 St. Gallen, Switzerland; joerg.neuweiler@kssg.ch

**Keywords:** kidney injury, alport syndrome, modifier gene, nephrin, podocin, glomerular basement membrane, slit diaphragm, focal segmental glomerulosclerosis

## Abstract

Kidney injury due to focal segmental glomerulosclerosis (FSGS) is the most common primary glomerular disorder causing end-stage renal disease. Homozygous mutations in either glomerular basement membrane or slit diaphragm genes cause early renal failure. Heterozygous carriers develop renal symptoms late, if at all. In contrast to mutations in slit diaphragm genes, hetero- or hemizygous mutations in the X-chromosomal *COL4A5* Alport gene have not yet been recognized as a major cause of kidney injury by FSGS. We identified cases of FSGS that were unexpectedly diagnosed: In addition to mutations in the X-chromosomal *COL4A5* type IV collagen gene, nephrin and podocin polymorphisms aggravated kidney damage, leading to FSGS with ruptures of the basement membrane in a toddler and early renal failure in heterozygous girls. The results of our case series study suggest a synergistic role for genes encoding basement membrane and slit diaphragm proteins as a cause of kidney injury due to FSGS. Our results demonstrate that the molecular genetics of different players in the glomerular filtration barrier can be used to evaluate causes of kidney injury. Given the high frequency of X-chromosomal carriers of Alport genes, the analysis of genes involved in the organization of podocyte architecture, the glomerular basement membrane, and the slit diaphragm will further improve our understanding of the pathogenesis of FSGS and guide prognosis of and therapy for hereditary glomerular kidney diseases.

## 1. Introduction

Focal segmental glomerulosclerosis (FSGS) is the most common primary glomerular disorder causing end-stage renal disease (ESRD) in the United States [1]. Podocyte injury plays a critical role in the pathogenesis of proteinuric kidney diseases, as podocytes are important for maintaining the glomerular filtration barrier [2]. They restore crucial components of the glomerular basement membrane (GBM), such as α3/α4/α5 type IV collagen chains (*COL4A3/4/5*), and of the slit diaphragm, such as nephrin (*NPHS1*) and podocin (*NPHS2*). Homozygous or hemizygous mutations in glomerular filtration barrier genes, such as the *COL4A3/4/5* genes, result in Alport syndrome (AS) [3,4,5], while homozygous mutations in the *NPHS1* and *NPHS2* genes result in congenital nephrotic syndrome [6,7]. The development of these syndromes leads to early ESRD. Few heterozygous carriers develop late changes, and they rarely (or never, in the case of *NPHS1/2* heterozygotes) develop ESRD. Polymorphisms in these genes are thought to result in an even milder phenotype or no phenotype.

The typical clinical signs of FSGS are marked proteinuria and podocyte injury. FSGS often manifests as nephrotic syndrome and frequently leads to renal failure. As FSGS is much less responsive to steroid therapy than minimal change disease, its prognosis for preserving renal function is (much) worse, with a high recurrence rate after renal transplantation [8]. 

The initial injuries leading to FSGS vary widely from monogenetic forms to secondary forms, which can be triggered by maladaptation of the podocyte to hyperfiltration, virus infections, drug use, or (unknown) circulating factors [8]. Primary (monogenetic) FSGS is caused by variants in the structural genes of the podocyte or the extracellular matrix (GBM). Primary FSGS typically results in in early onset of disease during childhood or adolescence.

Approximately 80% of adult cases of FSGS are primary (idiopathic) [1]. Up to 10% of familial FSGS can be explained by autosomal *COL4A3/4* mutations [9]; however, *COL4A5* mutations leading to X-linked Alport syndrome (XLAS) are much more common. Here, we describe three families with FSGS, which was unexpectedly diagnosed in toddlers with XLAS and in adolescent XLAS carriers with renal failure. In addition to mutations in the XLAS-related *COL4A5* gene, nephrin and podocin polymorphisms seem to have aggravated kidney damage, including severe FSGS with GBM ruptures in a toddler and unusually early renal failure in heterozygous girls.

## 2. Results

Patient 1 (case 1) was the index patient, in whom a severe kidney phenotype and GBM ruptures led to the discovery that in FSGS due to genetic GBM diseases, such as AS, polymorphisms in slit diaphragm genes can aggravate kidney damage. Subsequently, two other families with X-chromosomal AS were found to have FSGS, which was aggravated by slit diaphragm gene polymorphisms (Table 1).

### 2.1. Clinical Presentation

Patient 1 was a 27-month-old boy with persistent macrohematuria, proteinuria (1300 mg/L), active sediment, and normal renal function. His older sister and his non-consanguineous Lithuanian parents were healthy, with no family history of kidney diseases (Figure 1a). Post-infectious glomerulonephritis was excluded. Due to the initial suspicion of an infection and normal renal morphology on ultrasound examination, a cystoscopy was performed, which revealed hemorrhagic cystitis. However, common causes of hemorrhagic cystitis in childhood [10,11], such as cytomegalovirus or BK-polyomavirus infection, were ruled out. Consequently, a renal biopsy was performed. Light microscopy and immunohistochemistry (Figure 1b,c) revealed profound FSGS, IgM-positive deposits, and slight mesangial expansion. Ultrastructurally, the GBM presented with diffuse splitting, thinning, and ruptures (Figure 1d–f). The podocytes showed foot process effacement, with partial loss of the slit diaphragm (Figure 1d). These structural changes led to the diagnosis of AS. Hearing and eye evaluations did not reveal any abnormalities. Nephroprotective angiotensin-converting enzyme (ACE)-inhibitor therapy with ramipril was started [12,13], and the proteinuria slowly decreased from 1300 mg/L to less than 400 mg/L (Figure 1g). No further macrohematuria was reported.

Case 2 was an Austrian family with severe kidney disease in the mother, daughter, and son (Figure 2). The mother presented with hematuria and proteinuria in childhood. She soon developed ESRD and received a kidney transplant from her father at the age of 15. Her kidney biopsy at 11 years of age revealed advanced FSGS, hyalinosis, tubulointerstitial foam cells, podocyte effacement, splitting, lamellations, and partial thinning of the GBM. AS was considered; however, due to the unusual nature of a severe manifestation in a girl with healthy parents, she was diagnosed as having FSGS and nephrotic syndrome, with secondary structural changes in the GBM (Figure 2b,c). 

Her daughter and son both presented with hematuria and progressive proteinuria during the first year of life. A kidney biopsy performed when the daughter was two years old appeared relatively normal under light microscopy, although ultrastructural analysis showed advanced pathology, with podocyte effacement and splitting and thinning of the GBM, similar to the mother’s biopsy (Figure 2d–g). Hearing and eye evaluations did not reveal any pathological abnormalities in any family member. The family refused therapeutic intervention, leading to progressive proteinuria in both siblings (Figure 2h). Currently, the daughter and son have reached stage 3 of chronic kidney disease.

Case 3 was a family originating from Austria and Poland (Figure 3a). A kidney biopsy performed in a 27-year-old female (II-3) with hematuria and proteinuria showed advanced FSGS (Figure 3b,c). The biopsied material was not sufficient for ultrastructural analysis. A definite diagnosis was not possible; however, FSGS was suspected. The patient developed ESRD at approximately 40 years of age and received a kidney transplant at 51 years of age. Kidney disease progressed more slowly in her sister (II-7), who first presented with microhematuria at the age of 40. Further evaluation of several other affected family members (Figure 3a) revealed moderate hearing loss in patients II-3 and II-4 and eye involvement in patient II-7. Therefore, a complex hereditary kidney disease was considered.

### 2.2. Genetic Analyses of COL4A3/4/5 GBM Genes and NPHS1/2 Slit Diaphragm Genes

In case 1, the suspected diagnosis of AS was confirmed by genetic testing. A hemizygous X-chromosomal *COL4A5* de novo mutation was discovered in the boy but not in his parents: p.W1538X (TGG>TGA), c.4614 G>A. *COL4A5* mutations are typically associated with microhematuria in the first few years of life, slowly progressing to microalbuminuria below 300 mg [14], but not with high levels of proteinuria such as the 1300 mg/L observed in the two-year-old boy. Therefore, the severe phenotype of the two-year-old boy, an origin close to Scandinavia, and unusually severe FSGS led to the exploration of the slit diaphragm genes *NPHS1* and *NPHS2*. A polymorphism (p.R408Q (CGG>CAG), c.1223G>A) in *NPHS1* (nephrin) was identified in the boy and in his mother, who was not affected by the *COL4A5* mutation. Furthermore, homozygous silent polymorphisms in *NPHS1* (p.S1105S (TCG>TCA), c.102A>G) and *NPHS2* (p.G34G (GGA>GGG) c.3315G>A) were discovered in the boy (Table 1).

In case 2, a *COL4A5* splice mutation (exon 49, codon 1510, IVS49+3A>G) was hemizygous in the boy and heterozygous in the girl and the mother. No other family members were affected, suggesting that this was a de novo mutation in the mother (Figure 2 and Table 1). The heterozygous mother and daughter were both severely affected; however, proteinuria is usually mild or absent in young heterozygous *COL4A5* females, and early progression to ESRD before the third decade of life has never been described [14,15]. As in case 1, the severe phenotype of the heterozygous carriers led to an exploration of the slit diaphragm genes. Both heterozygous females, but not the hemizygous son, had a heterozygous *NPHS2* polymorphism (p.R229Q (CGA>CAA), c.686G>A; Table 1) that has been described as a modifier gene in thin basement membrane disease [16,17]. The clinical course of the son, who lacked the *NPHS2* polymorphism, was consistent with X-chromosomal AS in males. In contrast, the clinical courses of his heterozygous sister and mother, who had the *NPHS2* polymorphism, were very unusual.

In case 3, patient II-3 had a heterozygous *COL4A5* mutation (p.G624D (GGT>GAT), c.1871 G>A) that generally results in benign hematuria in females and late-onset ESRD in hemizygous males (in the 4th decade of life) [16]. Again, the severe phenotype resulted in further evaluation, and a polymorphism in *NPHS2* (p.R229Q (CGA>CAA), c.686G>A) in addition to a silent single-nucleotide polymorphism (SNP) in *NPHS2* (p.G34G (GGA>GGG), c.102A>G) were identified. Importantly, the G34G variant was found independently in case 1 and case 3, but represents a polymorphism of unknown relevance. All other family members presented only the *COL4A5* mutation (Figure 3a), and their clinical courses were consistent with benign hematuria (in females) and slowly progressing renal disease (in males) (Table 1). The heterozygous female with the *NPHS2* polymorphism (II-3) had higher proteinuria compared with her heterozygous sisters (II-7 and II-4) at the same age, who did not have the *NPHS2* polymorphism (Figure 3d).

### 2.3. Slit Diaphragm Gene Polymorphisms Aggravate Glomerular Architecture towards FSGS in Patients with GBM Mutations

Morphological analysis of case 1 further underscored the evidence that a *COL4A5* mutation in a basement membrane component can be aggravated by polymorphisms in slit diaphragm genes: (1) pronounced FSGS has not been previously described in toddlers with AS (Figure 1b,c); however, (2) initial thinning (and splitting) of the GBM is a common feature in children with AS (Figure 1d). GBM ruptures have not been previously described in patients with AS (Figure 1e). (3) Gross broadening of the podocyte foot processes, with partial loss of the slit diaphragm (Figure 1d), is a very notable and unusual finding in toddlers with AS, and (4) a high level of proteinuria (1300 mg/l with persistent macrohematuria) is unusual for a toddler with AS. The last feature and the FSGS were judged as being indicative of additional podocyte pathology. The structural changes led to the presumptive diagnosis of AS, though with the differential diagnosis of mutations in podocyte genes, as the atypically progressive structural changes could not be attributed to the patient’s age and his primary diagnosis of AS.

In case 2, the mother was diagnosed in childhood as having nephrotic-range FSGS with secondary GBM changes (Figure 2b,c). Kidney pathology in the 3-year-old daughter, with gross broadening of the podocyte foot processes (Figure 2f,g), was similar to that of her mother. In both individuals, despite their “merely” heterozygous AS status, the *NPHS2* polymorphism aggravated GBM pathology toward FSGS and early ESRD. Despite X-inactivation, ESRD has not been previously described in heterozygous AS carriers during adolescence. Notably, the hemizygous son, who had full-pattern AS but lacked the *NPHS2* polymorphism, had a clinical course that was identical to that of his “merely” heterozygous sister with the *NPHS2* polymorphism.

In case 3, the kidney pathology of the *COL4A5* carrier (II-3) was dominated by sclerosis (Figure 3b,c). This finding is a clear contradiction to the thin GBM that one would have expected in a young female with a heterozygous p.G624D mutation. This glomerulosclerosis may be due to the *NPHS2* polymorphism, as proteinuria was much lower in all other p.G624D-affected family members without the *NPHS2* polymorphism (e.g., the sister (II-4) at the same age). Even the *COL4A5* hemizygous adult male presented with less than 20% of the proteinuria of his aunt (II-3).

## 3. Discussion

It has been predicted that next-generation sequencing will identify most of the monogenic disease-causing genes by the year 2020 [18]. Next-generation sequencing has recently been shown to improve mutation screening in familial hematuric nephropathies [19]. However, the increasing knowledge of human genetic pathology will be associated with major challenges. Here, we show that for personalized medical care, physicians need all (1) a careful clinical evaluation of their patients, (2) close cooperation with the pathologist interpreting (kidney) specimens, and (3) genetic information about mutations and polymorphisms that might influence disease. Remarkably, thorough clinical exploration and its correlation with the kidney histology by the pathologist were essential tools guiding disease-focused genetic evaluation in the three families that we described. Without clinical and histological assessment preceding mutation analysis, physicians would not be able to understand and interpret the pathology of the glomerular filtration barrier [20].

As a limitation of our study, phenotyping of all family members and the extension of genetic correlation to other slit diaphragm genes was restricted by regional barriers, as family members originated from four different countries. The use of Sanger sequencing limited our search for possible disease-causing mutations to only five genes, whereas mutations in other podocyte genes could be contributory [21]. Additional regional legal barriers hindered us from extending the genetic analysis beyond the *COL4A3/4/5* and *NPHS1/2* genes. Still, our study was able to correlate the phenotype to the genetic changes in most members of a large three-generation pedigree, with a less severe phenotype in patients II-4, II-7, and III-7 without polymorphisms in slit diaphragm genes (Figure 3). 

Podocytes evolve into crucial cells that maintain the glomerular filtration barrier in renal diseases. Because of their need to withstand permanent filtration pressure, these cells adhere tightly to the underlying GBM [22]. The dynamic control of their cytoskeleton is affected by the slit diaphragm [22]. As a consequence, podocyte dedifferentiation, effacement, and FSGS are very common features in renal diseases [23]. Reducing the filtration pressure and thus protecting podocytes is a crucial therapeutic goal, even in children with congenital renal diseases [24].

Malone and coworkers demonstrated that up to 10% of familial FSGS can be explained by autosomal *COL4A3/4* mutations [9]. Our investigations expand previous findings that FSGS can also be caused by *COL4A5* mutations, the most common cause of AS, aggravated by polymorphisms in slit diaphragm genes [20,25]. α3/α4/α5 type IV collagen chains stabilize the GBM against filtration pressure [26]. Loss of any of the α3/α4/α5 type IV collagen chains results in a weaker GBM and increased podocyte cytoskeleton vulnerability [27]. As the cell–cell adhesions of the slit diaphragm are closely linked to the podocyte cytoskeleton, any polymorphism in slit diaphragm genes [28,29,30,31], as demonstrated in our study, might result in further damage in the histological picture of FSGS. This hypothesis should be tested in animal models using heterozygous *NPHS2*^+/R140Q^ mice, which correspond to the most common p.R138Q mutation found in humans and *COL4A3*^+/−^ mice [32,33,34,35]. *NPHS2*^+/R140Q^ mice develop no phenotype [32], while *COL4A3*^+/−^ mice develop benign familial hematuria [33,34]. The mature GBM with the α3/4/5 (IV) collagen chains can solely be built by podocytes, which sense the integrity of the GBM via their type IV collagen receptors, such as discoidin receptor 1 (DDR1) and integrins [4]. On the other hand, the crosstalk between the collagen receptor and the podocyte actin cytoskeleton, which also interacts with the slit diaphragm, might be the crucial link between the GBM and slit diaphragm that causes FSGS in our patients [4].

In conclusion, our findings demonstrate that after thorough clinical evaluation, the molecular genetics of different players in the glomerular filtration barrier can be used to evaluate FSGS. The analysis of genes involved in the organization of podocyte architecture, the GBM, and the slit diaphragm will further our understanding of the histological picture of FSGS. Our increasing knowledge of genes, arising from next-generation sequencing, will help to personalize the diagnosis and prognosis of and therapy for glomerular kidney diseases. Experienced physicians and pathologists are still needed to classify genetic results for the benefit of the patient.

## 4. Methods

### 4.1. Ethical Considerations

Written informed consent was given by the families for the publication of their cases. ICH-GCP data acquisition and storage of the European Alport registry was approved by the IRB of the University Medicine Goettingen, Germany (AZ 10/11/06; updated version from 2014). The European Alport registry has been registered at ClinicalTrials.gov (NCT 02378805) and as EudraCT number 2014-003533-25. 

### 4.2. Genetic Analyses

EDTA-treated blood was obtained after written informed consent was provided. Genomic DNA was extracted from peripheral blood leukocytes. The coding regions of *COL4A3*, *COL4A4*, *COL4A5*, *NPHS1*, and *NPHS2* were analyzed: The entire coding regions and splice sites of the *NPHS1* (transcript NM_004646.3; 29 exons), *NPHS2* (transcript NM_014625.2; 8 exons), *COL4A3* (transcript NM_000091.4; 52 exons), *COL4A4* (transcript NM_000092.4; 48 exons), and *COL4A5* (transcript NM_033380.2; 53 exons) genes were sequenced using unidirectional tagged primer Sanger sequencing in a 96-well format. Polymerase chain reaction (PCR) primers were designed using Primer-BLAST^®^ from the NCBI (National Center for Biotechnology Information, Bethesda, MD, USA). PCR products were generated using Buffer HOT Fire Polymerase^®^ (Solis Biodyne Solis, Tartu, Estonia) at an annealing temperature of 60 °C and were cleaned up using an Illustra ExoProStar One-Step^®^ kit (GE Healthcare, Freiburg, Germany). Sequencing was performed using a BigDye^®^ Terminator v1.1 cycle kit (Applied Biosystems^®^, Life Technologies GmbH, Darmstadt, Germany), and products were cleaned up using a CentriSep ABI^®^ kit (Applied Biosystems^®^, Life Technologies GmbH, Darmstadt, Germany). The sequences were read with an ABI Prism^®^ 3130XL DNA analyzer (Applied Biosystems^®^, Life Technologies GmbH, Darmstadt, Germany) and were analyzed using our in-house sequence analysis software (SeqEdit).

### 4.3. Morphological Analyses

Kidney biopsies were analyzed by light microscopy, immunohistochemistry, and electron microscopy. 

## Figures and Tables

**Figure 1 ijms-20-00519-f001:**
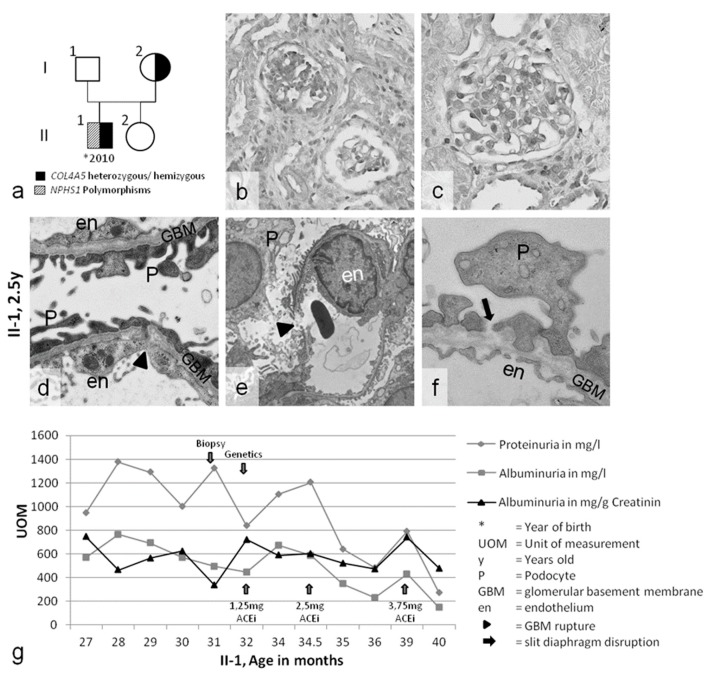
(**a**) Pedigree of case 1. (**b**–**f**) Nephropathological evaluation of the kidney biopsy of patient II-1. (**b**,**c**) Light microscopy showing focal segmental glomerulosclerosis (FSGS) and slight mesangial matrix expansion. (**d**–**f**) Ultrastructural analysis showing gross broadening of the podocyte foot processes; partial loss of the slit diaphragm (black arrow); and splitting, thinning, and ruptures of the glomerular basement membrane (GBM) (arrowhead). (**g**) The course of disease during ACE-inhibitor therapy: proteinuria constantly decreased (arrow). The diagnostic timescale is indicated by the blue arrows. Magnification: (**b**,**c**) 400×, (**d**) 20,000×, (**e**) 8000×, (**f**) 25,000×.

**Figure 2 ijms-20-00519-f002:**
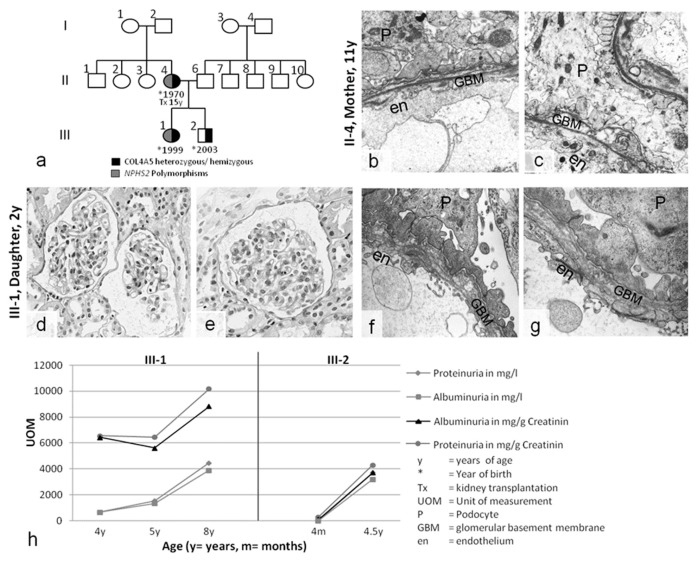
(**a**) Pedigree of case 2. (**b**,**c**) Kidney biopsy of the mother (11 y) revealing partial GBM thinning, splitting, and laminations in the lamina densa, with plump podocyte (P) foot processes. (**d**–**g**) Kidney biopsy of the daughter (3 y). (**d**,**e**) Light microscopy showing relatively normal glomerular and tubulointerstitial structures. Electron microscopy uncovered GBM pathology with splitting and thinning, similar to the mother’s nephropathology. (**h**) The course of disease without therapy in this X-chromosomal *COL4A5* genotype was unexpectedly very similar in the heterozygous girl (III-1) and her hemizygous brother (III-2): proteinuria constantly increased into the nephrotic range. Magnification: (**b**,**c**) 10,000×, (**d**,**e**) 400×, (**f**,**g**) 12,500×.

**Figure 3 ijms-20-00519-f003:**
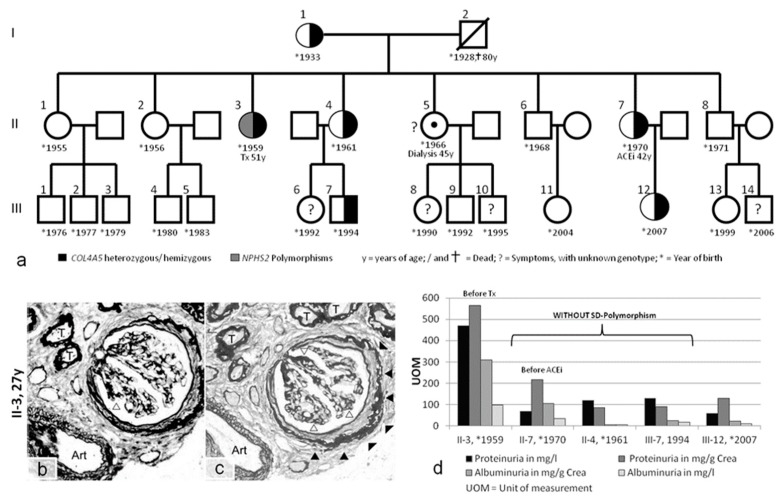
(**a**) Pedigree of case 3, with I-1 and family members (II-1, II-2, II-5, II-6, and II-8) living in Poland. Further evaluation was performed on family members living in Vienna (II-3, II-4, II-7, III-7, and III-12). Tx = kidney transplant. (**b**,**c**) Kidney biopsy in case II-3. Only a small core of renal tissue could be obtained for light microscopy. This tissue contained only one relatively intact glomerulus and three glomerular scars. The pathology was described as nonspecific and dominated by sclerosis. The glomerulus showed segmental scarring of the capillary loops (open arrowheads) and pronounced periglomerular fibrosis (solid arrowheads). The tubules (T) were dissociated by interstitial fibrosis, and thickening of the tubular basement membranes confirmed advanced atrophy. One small artery (Art) exhibited minimal intimal fibrosis. (**d**) Proteinuria and albuminuria in family members with and without additional slit diaphragm (SD) polymorphisms. Methenamine silver (**b**) and PAS staining (**c**); magnification: 400×.

**Table 1 ijms-20-00519-t001:** Summary of patient phenotypes and genotypes.

Patient	Sex	First Clinical Presentation > Symptoms	Genotype *COL4A5*	Genotype *Nphs-1/-2*	Ear/Eye	Dialysis/Tx	Medication	Affected Family Members
**Case 1** **A.N. II-1**	♂	27 months macrohematuria acanthocytes proteinuria urinary tract infection	*COL4A5* p.W1538X (TGG>TGA) hemizygous	*Nphs1*: p.R408Q (CGG>CAG) polymorphism heterozygous *Nphs1*:p.S1105S(TCG>TCA) polymorphism homozygous *Nphs2*: p.G34G (GGA>GGG) polymorphism homozygous	no pathological findings	none	2012–today ACEi	no kidney diseases known mother without symptoms: *Nphs1*: p.R408Q (CGG>CAG) polymorphism heterozygous
**Case 2** **C.V. II-4**	♀	11 years macrohematuria proteinuria urinary tract infection	*COL4A5* Exon 49, Codon 1510, IVS49+3A>G heterozygous	*Nphs2*: p.R229Q (CGA>CAA) polymorphism heterozygous	no pathological findings	Tx age: 15	/refused treatment	no other family members affected
**S.V. III-1**	♀	1 year macrohematuria Proteinuria	*COL4A5* Exon 49, Codon 1510, IVS49+3A>G heterozygous	*Nphs2*: p.R229Q (CGA>CAA) polymorphism heterozygous	no pathological findings	none	/refused treatment	
**M.V. III-2**	♂	4 months macrohematuria proteinuria	*COL4A5* Exon 49, Codon 1510, IVS49+3A>G hemizygous	none	no pathological findings	none	/refuse treatment	
**Case 3** **L.U. II-3**	♀	27 years hematuria proteinuria	*COL4A5* p.G624D (GGT>GAT) heterozygous	*Nphs2*: p.R229Q (CGA>CAA) polymorphism heterozygous *Nphs2*: p.G34G(GGA>GGG) polymorphism homozygous	minimal high-frequency hearing loss (2013)	Tx age: 51	Before Tx: ACEi from 33 years	see pedigree of family (Figure 3)
**W.T. II-7**	♀	40 years: microhematuria 42 years: proteinuria	*COL4A5* p.G624D (GGT>GAT) heterozygous	none	retinal detachment	none	02/2013: ACEi (discontinued due to angioedema)	
**T.O. II.4**	♀	microhematuria	*COL4A5* p.G624D (GGT>GAT) heterozygous	none	high-frequency hearing loss (2013)	none	none	
**S.O. III-7**	♂	microhematuria	*COL4A5* p.G624D (GGT>GAT) heterozygous	none	no pathological findings (2013)	none	none	
**O.T. III-12**	♀	no symptoms	*COL4A5* p.G624D (GGT>GAT) heterozygous	not investigated	not investigated	none	none	

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
