# Peer review of "Kidney Injury by Variants in the COL4A5 Gene Aggravated by Polymorphisms in Slit Diaphragm Genes Causes Focal Segmental Glomerulosclerosis"

_ijms, 2019, doi:10.3390/ijms20030519_

Round 1
Reviewer 1 Report
The idea underlying the work by Frese and colleagues is original, the Authors aim to unveil a mechanism involved in focal segmental glomerulosclerosis which is a crucial disorder leading to end-stage renal disease.
Thus the manuscript is interesting, and also well written.
I just would like to suggest a couple of minor revisions that could improve the work prior to publication.
In the introduction section:
- some more information about FSGS as well as about podocyte injury (when and how they occur and which are the major clinical signs) should be provided
- how podocytes restore crucial components of the GBM and of the slit-diaphragm should be clarified.
Author Response
The idea underlying the work by Frese and colleagues is original, the Authors aim to unveil a mechanism involved in focal segmental glomerulosclerosis which is a crucial disorder leading to end-stage renal disease.
Thus the manuscript is interesting, and also well written.
I just would like to suggest a couple of minor revisions that could improve the work prior to publication.
In the introduction section:
1) some more information about FSGS as well as about podocyte injury (when and how they occur and which are the major clinical signs) should be provided
Response: Thank you for the suggestions. We now added a paragraph explaining FSGS and which the major clinical signs are. In addition, when and how podocyte injury can cause FSGS is explained on page 2, second paragraph of the introduction section.
2) how podocytes restore crucial components of the GBM and of the slit-diaphragm should be clarified.
Response: We agree. We added the podocyte function (GBM and slit-diaphragm) on page 2 as third paragraph.
Reviewer 2 Report
The present investigation describes the possible impact of mutations in the COLA45 gene in conjunction with distinct polymorphisms of the nephrin and podocin genes in view of FSGS phenotype.The issue is very interesting since it shows the impact of multiple gene alterations for a clinical important entity. FSGS /proteinurie seem to be determined by different compartment of the glomerulus, and does not originates exclusively from “podocytopathies”. The manuscript is well written and substantiated by genetic and histological analysis, including electron microscopy for ultrastructural analysis which is clearly valorizing. The data are relevant for clinical practice.
-The data here are obtained from 3 cases, yet representing a case series. The “case series character” of the study should be indicated either in the headline or in the abstract.
-The authors should comment and describe a little more in detail how the mutations in both genes “work together” in the pathological sense.
- Is it really the case that a COLA5gene mutation is aggravated by e.g. a nephrin gene polymorphism? Isn`t possible that e.g. pathologic nephrin gene effects are aggravated by COLA5 gene changes on the other hand? This should be issued in the discussion section.
Author Response
The present investigation describes the possible impact of mutations in the COLA45 gene in conjunction with distinct polymorphisms of the nephrin and podocin genes in view of FSGS phenotype. The issue is very interesting since it shows the impact of multiple gene alterations for a clinical important entity. FSGS /proteinurie seem to be determined by different compartment of the glomerulus, and does not originates exclusively from “podocytopathies”. The manuscript is well written and substantiated by genetic and histological analysis, including electron microscopy for ultrastructural analysis which is clearly valorizing. The data are relevant for clinical practice.
1) The data here are obtained from 3 cases, yet representing a case series.The “case series character” of the study should be indicated either in the headline or in the abstract.
Response: We agree. We clarified the case series character in the abstract on page 1.
2) The authors should comment and describe a little more in detail how the mutations in both genes “work together” in the pathological sense.
Response: Thank you for the suggestions (which are similar to point 1 and 2 of reviewer 1).We added the new paragraphs 2 and 3 on page 2 in the introductions section, which explain how the mutations in both genes affect the GBM/slit-diaphragm. In addition we comment on, how the both genes might “work together” in 5th paragraph of the discussion section on page 8.
3) Is it really the case that a COLA5gene mutation is aggravated by e.g. a nephrin gene polymorphism? Isn`t possible that e.g. pathologic nephrin gene effects are aggravated by COLA5 gene changes on the other hand? This should be issued in the discussion section.
Response: We agree. We discuss this in the same paragraph 5 of the discussion section on page 8.
Reviewer 3 Report
This is a manuscript by Frese et al on the effects of nephrin and podocine polymorphisms on FSGS/AS. This is nice work and can be improved further. Here are my comments:
1. Institutional approval must be obtained (Ethical concern) in addition to patient consent.
2. Proteinuria and albuminuria should be presented as mg/mg or mg/g creatinine. mg/L is protein concentration and protein concentration can vary based on urine concentration.
3. The author claims of widespread glomerulosclerosis should be supported by including more glomeruli, 2 glomeruli, 1 of which has sclerosis does not support "widespread" (Figure 1)
4. Polymorphisms should be noted by "rs" numbers
5. The authors should describe the effect of polymorphisms better i.e. what is the evidence and why they think the polymorphisms would exacerbate the phenotype, otherwise it is very speculative. Include references J. Clin. Invest. 110: 1659-1666, 2002. [PubMed: 12464671], Am. J. Hum. Genet. 89: 139-147, 2011. [PubMed: 21722858], Nature Genet. 46: 299-304, 2014. [PubMed: 24509478], Pediat. Nephrol. 28: 2061-2064, 2013. [PubMed: 23800802]. I did not find references for G34G. In addition, the authors should include similar evidence for NPHS1 polymorphism
Author Response
This is a manuscript by Frese et al on the effects of nephrin and podocine polymorphisms on FSGS/AS. This is nice work and can be improved further. Here are my comments:
1. Institutional approval must be obtained (Ethical concern) in addition to patient consent.
Response: Thank you for this comment and please excuse that we did not include this information right in the beginning. We now provide the IRB-approval (AZ 10/11/16) and ClinicalTrials.gov/EudraCT number of the European Alport registry on page 9.
2. Proteinuria and albuminuria should be presented as mg/mg or mg/g creatinine. mg/L is protein concentration and protein concentration can vary based on urine concentration.
Response: We agree and apologize for the different concentrations and units, which are caused by very different lab reports from family members of 4 different European countries and more than 10 different primary physicians. Whenever possible, we use mg/mg or (preferred) mg/gCrea.
3. The author claims of widespread glomerulosclerosis should be supported by including more glomeruli, 2 glomeruli, 1 of which has sclerosis does not support "widespread" (Figure 1)
Response: we agree and deleted “widespread” from the text on page 4, 1st paragraph.
4. Polymorphisms should be noted by "rs" numbers
Response: We understand that we provide different details about the variants throughout the text of the specific figures, which might be misleading. Similar to your point 2, this is caused by different labs from different countries. The most detailed information about the variants including the polymorphisms is provided in section 2.2 “Genetic analyses of COL4A3/4/5 GBM genes and NPHS1/2 slit-diaphragm genes” on page 6.
5. The authors should describe the effect of polymorphisms better i.e. what is the evidence and why they think the polymorphisms would exacerbate the phenotype, otherwise it is very speculative. Include references J. Clin. Invest. 110: 1659-1666, 2002. [PubMed: 12464671], Am. J. Hum. Genet. 89: 139-147, 2011. [PubMed: 21722858], Nature Genet. 46: 299-304, 2014. [PubMed: 24509478], Pediat. Nephrol. 28: 2061-2064, 2013. [PubMed: 23800802]. I did not find references for G34G. In addition, the authors should include similar evidence for NPHS1 polymorphism
Response: We apologize for this misunderstanding regarding G34G: we also have no reference and are aware of the unknown relevance, but we think that the G34G variant needs to be described in this context as we found it independently in case 1 and case 3. As a response, we clarify this in the text on page 6 in the first paragraph of the 2.2. section.
Thank you for the suggestions (which are similar to point 1 and 2 of reviewer 1). We added the new paragraphs 2 and 3 on page 2 in the introductions section, which explain how the mutations in both genes affect the GBM/slit-diaphragm. In addition we comment on, how the both genes might “work together” in 5th paragraph of the discussion section on page 8.
Round 2
Reviewer 3 Report
The authors addressed most of my suggestions although the manuscript can be improved further
Discussion about cause and effect needs to be better
Author Response
Dear reviewer 3,
thank you for the response. I understand your comment about further the cause and effect of our findings. However, evidence in this field is somewhat limited and rather speculative. Our current project "Role of the interaction of slit membrane, podocyte and glomerular basement membrane in pathogenesis of glomerular kidney diseases such as Alport's Syndrome" founded by the German Reserach Foudation DFG (GR 1852/6-1) addresses your wish for further discussion of cause and effect of the interctaion between the podocyte, GBM and slit-diaphragm.
Sincerely, Oliver Gross